# A Procedural World Generation Framework for Systematic Evaluation of Continual Learning

**Timm Hess, Martin Mundt, Iuliia Pliushch, Visvanathan Ramesh**
Goethe University, Frankfurt am Main, Germany
`hess@ccc.cs.uni-frankfurt.de`
`{mmundt, pliushch, vramesh}@em.uni-frankfurt.de`

## Abstract

Several families of continual learning techniques have been proposed to alleviate catastrophic interference in deep neural network training on non-stationary data. However, a comprehensive comparison and analysis of limitations remains largely open due to the inaccessibility to suitable datasets. Empirical examination not only varies immensely between individual works, it further currently relies on contrived composition of benchmarks through subdivision and concatenation of various prevalent static vision datasets. In this work, our goal is to bridge this gap by introducing a computer graphics simulation framework that repeatedly renders only upcoming urban scene fragments in an endless real-time procedural world generation process. At its core lies a modular parametric generative model with adaptable generative factors. The latter can be used to flexibly compose data streams, which significantly facilitates a detailed analysis and allows for effortless investigation of various continual learning schemes.

## 1 Introduction

In an era where deep neural networks have diffused into every conceivable application, a natural interest in the long-standing challenge of *catastrophic interference* [1, 2] in continuous training has resurfaced. Various families of approaches have emerged to alleviate this challenge of formerly encoded representations being rapidly superseded with the arrival of novel distinct data from a non iid data distribution continuum [2, 3, 4, 5, 6, 7, 8, 9, 10, 11, 12, 13, 14, 15]. However, despite the asserted progress, recent reviews repeatedly stress the importance of more exhaustive and realistic evaluation [16, 17, 18, 19, 20, 21, 22, 21, 23]. Notably in computer vision, the majority of presently emphasized benchmarks are contrived variants of the prevalent existing datasets [24, 25, 26, 27, 28, 29], where individual concepts of the datasets are split into disjoint subsets and presented to the learner in sequence, or deliberately designed to follow this trend of object and class increments [30, 31]. In hindsight, the latter benchmark construction neglects two imperative elements. First, it disregards a myriad of real-world continual learning scenarios, where the environment and its various conditions can be subject of constant change. Second, the uncontrolled data acquisition process hinders insights on a method's feasibility beyond the specific empirical outcome.

We posit that generation of synthetic data through the use of parametrized generative models can provide a remedy for the present lack of a more detailed continual learning analysis. Here, the crucial realization is that catastrophic forgetting is a consequence of dense entangled representations in neural networks being greedily overwritten by newly encountered information. As such, the catastrophic interference phenomenon is a result of chosen optimization strategies and likewise applies to any investigation of synthetic data. More so, we conjecture that: *if catastrophic forgetting cannot be circumvented in scenarios with a known synthetic data foundation, there is limited hope to understand limitations and overcome the challenge in real-world settings.*

35th Conference on Neural Information Processing Systems (NeurIPS 2021) Track on Datasets and Benchmarks.

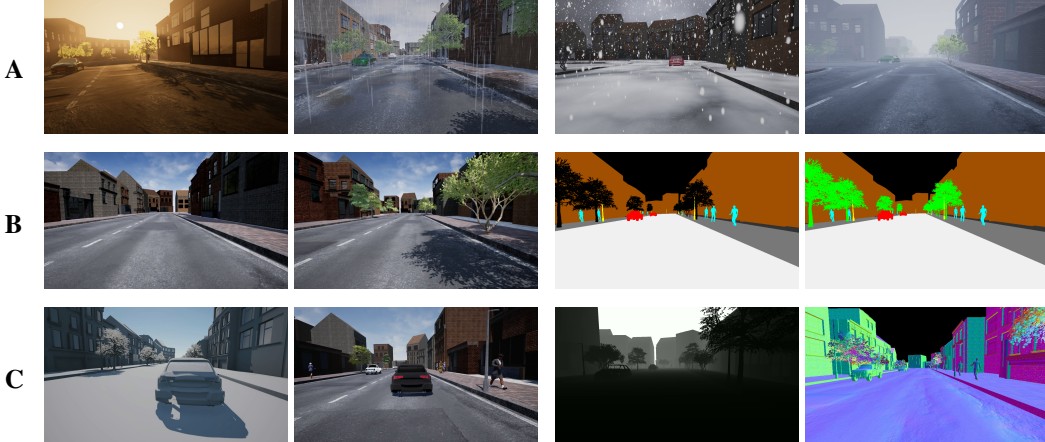

Figure 1: Example video stream snapshots. Row A illustrates common environmental changes, such as variations in illumination and weather conditions. Row B depicts two potential examples for class incremental learning, where entire object classes, here trees, progressively appear or disappear (left image pair), or the learning task is based on successive availability and granularity of annotations (right image pair). Row C shows exemplary de-activation of specific physics-based material properties such as color, surface normals, or (sub-)surface reflections, resulting in a fully gray or colored flat shaded world, without any reflection or small cavity details (left image pair). Built-in availability of additional depth and surface normal annotations is further highlighted (right image pair).

In principle, the idea to leverage virtual data has already found countless prior applications, primarily due to simulators' ability to yield automatic precise ground truth information [32, 33, 34, 35, 36, 37, 38, 39, 40, 41, 42]. Computer-graphics frameworks thus typically facilitate sampling of a maximum of conceivable variations to enable deep learning in domains where scares data is available and real-world data acquisition is insurmountable. However, in continual learning, scenarios of interest should inherently encompass knowledge about the detailed temporal shifts in the observed distribution. These range from occurrence of particular objects, their geometry and texture, the frequency and order of objects' (dis-)appearance in the scene, or continuous changes in the environmental weather and lighting. Corresponding investigations of continual learning thus require straightforward accessibility to meticulous control of the real-time online changes in the independent generative factors. With a focus set on large-scale annotated data generation to overcome an existing lack of data, the latter nuanced control is generally not exposed to the user in existing simulators' surface controls.

Inspired by previous works in the context of urban scene segmentation [43, 44, 45, 38], our primary contribution is the introduction of a modular Unreal Engine 4 [46] based 3-D computer graphics simulator that now also enables clear-cut generation and assessment of diverse continual learning scenarios. A selection of video snapshots is illustrated in figure 1 and additionally a showcase video can be viewed at `https://youtu.be/8zDhol8CIf0`. Specifically, we:

- Introduce a simulator that facilitates grounded investigation of continual learning mechanisms through access to highly customizable data. At all times, our simulator only renders an upcoming segment of the world through efficient real-time scene assembly. Its offered data generation is based on manipulation of temporal priors and parameters of the generative model. Our modular control spans aspects from physics-based (de-)activation of color, surface normals and scattering, to switches in weather conditions or environment lighting, and ultimately to commonly evaluated abrupt changes in the data population though (dis-)appearance of entire object categories.

- Corroborate our simulator's utility in an initial showcase of multiple continual learning techniques, investigated across incremental class, environmental lighting, and weather scenarios.

- Provide open access to:
  1. As a benchmark creation tool, a stand-alone simulator executable with configuration files for the specification of rendered sequences: `https://doi.org/10.5281/zenodo.4899294`
  2. To allow extensions, the underlying source-code of the simulator: `https://github.com/ccc-frankfurt/EndlessCL-Simulator-Source`

3. A set of respective videos and their precise dataset versions to reproduce the particular experiments of this paper: `https://doi.org/10.5281/zenodo.4899267`.

We have made use of the Zenodo platform to ensure persistence of our datasets and software, while also making sure that our content has a DOI with versioning capabilities for future updates.

## 2 Endless Procedural Driving Simulator

Our procedural world generation framework allows for creation of temporally consistent video streams, where respective sub-sequences are subject to an interpretable parametric generative model through which the scenario is continuously adaptable. This can be gradual and seamless changes in the environment, or mirror abrupt shifts in the world's configuration. Our specific implementation is inspired by urban driving. Our considered main actor is a *vehicle* with statically attached *camera* that drives along a procedurally generated track of successive *street segments*. Every such street segment is randomly selected to balance curvature and crossings with straight roads. For each sampled street element, various *objects*, such as *buildings*, *trees* or *street lamps* are stochastically placed according to real-world motivated priors. Additional *dynamic actors*, i.e. *other vehicles* and *humans* are sampled for each segment, their *motion* dynamics and *animations* drawn at random. At all times, the number of existing street segments remains constant. As the main camera actor proceeds through the world, novel segments are procedurally generated, while already observed ones are disposed of. Through control of spawning probabilities or parameters administering *weather* and *lighting*, the user is granted the ability to adapt the upcoming world of the real-time generated video.

### 2.1 Generative Model

We define an entire video sub-sequence as a random variable $X_t = \langle \text{S,B,Tr,Lp,H,V,C,W,L,E,R} \rangle$, with street segment S, static buildings B, trees Tr, and street lamps Lp, dynamic human and vehicle actors H and V, a car with attached camera C, weather condition W and lighting L. A meta-variable E governs the existence of entire object and actor categories and R controls the physics-based material rendering model. A continuously growing video is therefore comprised of $t = \{1, ..., T\}$ sub-sequences, defined by the parametrized probabilistic generative process given by:

$$
\begin{aligned}
P(X_t|\Theta) =& P(V_t|V_{t-1}, S_t, \theta_{V,t}, E_{V,t}, R_t)\, P_t(B, Lp, Tr|S, \theta_B, \theta_{Tr}, \theta_{Lp}, E_B, E_{Tr}, E_{Lp}, R) \\
& P(C_t|C_{t-1}, \theta_{C,t}, S_t, R_t)\, P_t(A|\theta_A, H)\, P_t(H|S, \theta_H, E_H, R) \\
& P_t(E_B, E_{Tr}, E_{Lp}, E_H, E_V|\theta_E)\, P_t(S|\theta_S, R) P_t(R|\theta_R) P_t(W|\theta_W)\, P_t(L_{\text{I}}, L_{\text{D}}|\theta_L).
\end{aligned}
\tag{1}
$$

Here, notation is simplified for $R = \{R_M, R_C, R_N, R_R\}$, manipulating a material's metallicity, color, surface normals and roughness, and $P_t$ is short for an index $t$ at every entry of an expression. All variables and their stochastic dependencies are illustrated in the graphical model of figure 2. We proceed by elaborating on the random variables and their distributions parametrized by respective $\Theta$.

### 2.2 Random Variables and Distributions

The random variables of our generative process are subject to three families of distributions: discrete categorical or Bernoulli, and continuous distributions on bounded intervals. The former two generally reflect a random selection of e.g. object or actor styles, and their general existence and co-occurrence. The latter expresses our belief in e.g. plausible object or actor locations through parametrization of the finite support and their overall amount. With few exceptions and as will be detailed in an instant, these variables are presently assumed to originate from uniform distributions, but can generally be sampled with more complex distributions and interdependencies, if for instance a specific city composition is desired to be emulated.

**Street segments, main actor and vehicles:** The overarching categorical random variable $S$ indicates the stochastic selection of street-segment layouts, i.e. elements containing various straight and curved road designs, or crossings, with respective sampling probability values given by $\theta_S$. The segments themselves are not placed stochastically. Instead, the beginnings of consecutive sequence elements are deterministically attached to their predecessor's end, in order to seamlessly continue the endless procedural generation of the world. The main actor of a vehicle with statically coupled camera, random variable $C$, follows this world as if it were on a pre-determined track, i.e. the direction of

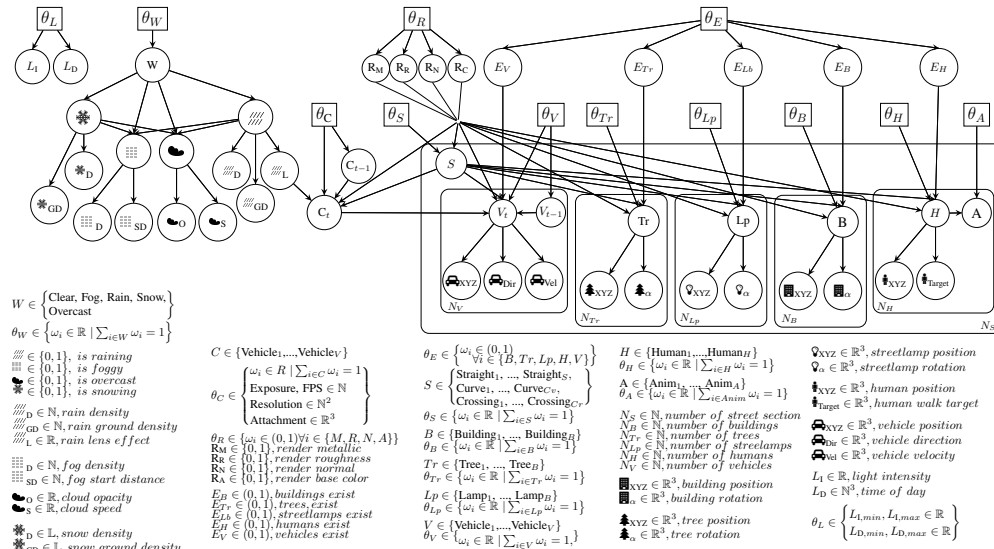

Figure 2: Parameterized probabilistic graphical model. Plate notation [47] is used for repeated random variables, sets of parameters $\Theta$ are denoted by rectangles. As the generative process creates entire video sub-sequences, only the camera $C_t$ and vehicle actors $V_t$ have an explicit dependency between two consecutive sampling steps. All other random variables are constrained to reside in their respective sub-sequence and consecutively sampled independently. The subscript $t$ has thus been omitted for ease of readability. The physics-based rendering random variables $R_{M,R,N,C}$ are each connected to all objects and actors. Their arrows have been merged visually to avoid excessive clutter.

the camera vehicle is thus contained in the stochastic sampling of street segments. $C$ itself is of categorical nature and represents a choice in vehicle model, with additional $\theta_C$ describing camera properties such as resolution, exposure or the captured video's frames per second. In principle, it is not directly subject to a random velocity. However, the velocity does vary in dependency on other stochastic variables. These include further dynamic vehicles, expressed through the categorical variable $V$ and its parameters $\theta_V$. Again, these vehicles are randomly chosen from the repertoire of vehicle models. Their location is sampled from a uniform distribution on a bounded interval that is constrained to the extent of the sampled street segment, their velocity selected at uniform in the range of assumed minimally and maximally realistic velocities in an urban scenario. If these vehicles drive slowly in front of the camera main actor vehicle, they inevitably lead to deceleration in order to avoid collision. As occasional auxiliary dynamic vehicles can thus continue their path and persist across multiple video sub-sequences, they represent the only random variable apart from the main camera that relies on its former sub-sequence's state.

**Static objects and human actors:** All other dynamic and static random variables are contained within their respective street segments. These include a population of possible static 3D-objects, presently captured by the set $Obj = \{B, Tr, Lp\}$, corresponding to buildings, trees and street lamps, and additional dynamic human actors $H$. Again, the respective categorical random variable indicates a choice in available models. Human actors are further dependent on categorical random variable $A$, enabling the random choice from a discrete set of animation. Once more, their sampling probability is given through respective parameters, and their stochastic initial and final pose, representing the walk target, is presently drawn from uniform distributions with finite support, determined by the scope of the street segment on which they are randomly placed.

**Category existence:** With the exception of the always present world's street segments and tracked camera vehicle, all previously introduced categorical random variables are modulated through Bernoulli variables on a top-level. The corresponding $E_i \; \forall i \in \{B, Tr, Lp, H, V\}$, parametrized by $\theta_E$, govern the overall "existence". In a sense this is a convenience random variable, that allows trivial control over the presence of entire categories, in foresight of class incremental scenarios where e.g. all trees appear or disappear when driving through or leaving an avenue. In general, this could have been adjusted through the number of spawned objects and actors, i.e. the repeated sampling of

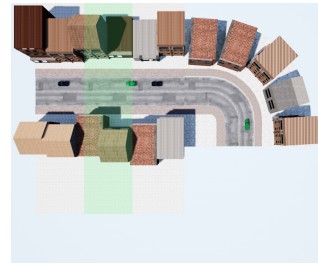 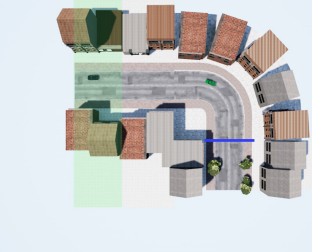 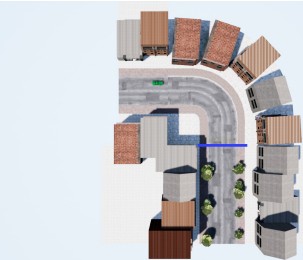

Sub-Sequence 1, Tile 4          Sub-Sequence 2. Tile 1          Sub-Sequence 2, Tile 2

Figure 3: Birds eye view of the stochastic procedural scene generation. Switching the scene's configuration from 'Sub-Sequence 1' to 'Sub-Sequence 2' it can be observed how the object population changes to exclude cars and feature trees in the newly added tiles. The area highlighted in green marks the extent of a single tile and the blue line indicates the seam between the two sub-sequences.

random variables for one sub-sequence expressed through plate notation [47] in figure 2. For ease of readability, we define the respective random variable $N_i \forall i \in \{B, Tr, Lp, H, V\}$ independently and for now assume that distinct amounts, limited by an empirical $N_{i,max}$, are a priori equally likely.

**Weather and lighting:** A complementary part of the graphical model controls weather and lighting. Categorical $W$, parametrized by $\theta_W$, represents an initially mutually exclusive choice between five distinct weather conditions: clear, fog, rain, snow and overcast (cloudy). Depending on the concrete outcome, an additional set of Bernoulli variables covers potential co-occurrence of fog and clouds when rain or snow are active. Each of these conditions is subject to further random variables on density, ground density (for snow cover and puddles), and camera lens effects. For now, these latter variables are sampled from uniform distributions ranging from zero to unity in terms of effect strength. In complete analogy, lighting for a video sub-sequence is given by continuous random variables on intensity and daytime, defining the illumination angle and color due to atmospheric scattering.

**Physically-based rendering:** Finally, the world is assumed to follow a physics-based rendering process. Parametrized by $\theta_R$, a set of Bernoulli random variables $R_r \forall r \in \{M, R, N, C\}$ dictates the activation of rendering aspects with respect to all materials, i.e. metallicity, surface roughness, normals, and base color. These define how material appearance manifests in the observed video-subsequence, with potential to exclude color or particular surface or subsurface light interactions.

## 2.3   Practical Details

The practical simulation is based on Unreal Engine 4 [46], supporting our requirements for specification of distributions through code and dynamic world generation, while aiming for real-time high photo-realism. The essential street segments are represented through a discrete set of tiles, where for each tile type spawn volumes govern the finite support bounds of the underlying location sampling for each object and actor. In addition, each such tile includes anchor points for attachment with further tiles. Figure 3 shows a birds eye perspective of a constructed world segment and illustrates the procedural scene generation process. For details on assets, distribution parameter choices, and the spawning bounds we refer to the appendix. The use of tiles elegantly encapsulates the generative model's probabilistic sampling properties in practice, while yielding considerable memory and computation benefit by strictly limiting the amount of resources to be managed at each time step.

We simultaneously capture color video, semantic pixel annotations, depth, and normals. Continuous world generation with capture of a video with half-HD ($960 \times 540$) resolution, from a single camera, achieves $\sim 15$ frames per second (fps) on a consumer GTX-1080 GPU. With only the original video and one additional mode, this number increases to $28$ fps. In either way, we allow the user to set a desired capturing rate and make use of time dilation, i.e. virtually slowing down the scene, if the same world segment is needed to be captured with higher frame rate or extensive rendering features such as above full-HD resolutions. As pointed out in our contributions, we provide a stand-alone simulator executable. Here, used content is encrypted and can be used without explicitly sharing licensed assets. Arbitrary chains of distribution modifications for video sequences to be generated can be specified through a JSON-config file. For inclusion of novel assets we point the reader to our shared source-code. We provide more detailed descriptions and usage instructions in the supplementary.

# 3 Related Work

The idea to leverage synthetic data creation for the training of deep neural networks has seen an impressive amount of successful practical implementations [32, 33, 34, 35, 36, 37, 38, 39, 40, 41, 42]. The common expectation is that learned appearance models can be adapted in their domain to an ultimately desired real world target task. Various works have therefore focused on automated calibration based on similarity computations between virtual and real images [33, 34], manifold alignment to match scale between generated and synthetic texture data [32], or adversarial tuning to assimilate data population statistics [48, 49]. On the alternative end of the respectively constructed scenes, the general aim is thus to augment real world datasets [35], customize aspects that are otherwise difficult to capture, such as pedestrian motions [36] or human actions [40], in an effort to overcome laborious human annotation in creation of large-scale datasets with massive amounts of variation. Depending on specific works, the spotlight can be on e.g. fixed urban scenes and variations of environmental factors [44, 39], direct extraction of such scenes from video games [38], or the complete randomization of all factors to maximize the amount of conceivable configurations [37, 41, 48]. Whether or not all factors are randomized in stochastic processes or scene elements remain static, the generally measured utility and impact is derived from measurements on corresponding real world benchmark datasets [45, 27, 45, 29, 43].

Although valuable for their proposed purpose, we posit that existing simulators are not natural for exploratory study of continual learning limits. In simulators that use pre-determined layouts the user is limited to specific geometrical configuration and scene types. If assumed static buildings and other actors need to be replaced, removed or complete object categories are desired to be added, the world needs to essentially be recomposed [36, 38, 44, 39]. Analogously, works that employ stochastic point processes or similar hierarchical procedures to randomize entire scene configurations, including stochastic camera placement to vary frames or locally consistent video segments [44, 48, 50, 41, 40], require significant amounts of compute in recurring composition of the entire world if the user wishes to change priors on underlying generative factors. As an additional challenge this is only possible if, and only if, the associated simulator core and source code has been publicly released beyond an executable, because the desired settings are typically not applicable through the provided user interface. Our proposed simulator differentiates in this regard, as we can dynamically change parameters in real-time generation to e.g. spawn trees in the distance, de-spawn specific buildings, change locations, vary lighting and modulate weather effects. Such nuanced adaptability of all scene elements enables easy creation of a dynamically adaptable endless procedural world.

From the perspective of continual learning, our described simulator thus allows for composition of data sequences in an effort to extent presently limited analysis. In particular, we can enable analysis beyond present continual vision practice, that rests primarily on not well understood sequentialization of popular classification benchmarks [24, 25, 26, 27, 28, 29]. The latter typically undergo class specific splits, permutation, concatenation or other alterations through augmentation to provide pre-designed iterative sequences of object or class information [30, 31]. Based on these contrived and uncontrolled benchmarks multiple families of continual learning approaches have formed. These range from simple rehearsal of original data subsets [2, 3, 4, 5] to generative data replay [6, 7, 8], or from functional regularization [9, 10, 11], based on knowledge distillation [51], to explicitly constraining parameters [14, 13]. Unfortunately, in empirical comparison, it quickly becomes apparent that assumptions of individual methods are narrow and seem to often be practically tailored towards the limited use case of a particular benchmark [16, 17, 18, 19, 20, 22, 21, 23]. We would argue that this is not necessarily a direct result of originally misguided design, but rather a consequence of the underlying original datasets being seldomly designed with continual learning in mind. Our works imminent goal is thus to deepen our understanding of when and why deep learning fails in continuous training, how potential curricula impact learning, and how mechanisms can be improved to consistently mitigate shortcomings across a wider range of scenarios.

# 4 Deep Continual Learning Experiments

We empirically corroborate our simulator's utility in a set of initial experiments. Here, we showcase that catastrophic interference can still be a significant challenge across many literature methods, even when only considering simulated data. Inspired by the typically limited evaluation of deep continual learning in class incremental scenarios [16, 17, 18, 19, 20, 21, 22, 23], we now generate

and investigate video sequences in three distinct set-ups: *incremental class appearance*, *varying weather conditions* and *decreasing illumination intensity*. They have been selected to display the benefits and shortcomings of currently prevalent techniques to alleviate catastrophic interference, and consequently why it is necessary to make use of our simulator for a more diverse evaluation.

For this purpose we consider popular approaches from various families of continual learning mechanisms: synaptic intelligence (SI) [14] and elastic weight consolidation (EWC) [13] for parameter regularization, functional regularization through knowledge distillation as presented in learning without forgetting (LwF) [9], as well as data replay methods. For the latter, we consider replay using gradient episodic memory (GEM) [22], a straightforward exemplar rehearsal mechanism, where a subset of data is stored and interleaved in continuous training in the spirit of [52, 3], and generative pseudo-replay with open set classifying denoising variational auto-encoder (OCDVAE) [8].

### 4.1 An Initial Set of Considered Scenarios

For our investigation, we have selected the simplest conceivable task of classification, where all objects' bounding boxes are assumed to be detected perfectly. Even in this significantly facilitated setting we will see that many investigated techniques are more brittle than desired. We emphasize that our simulator is naturally capable of rendering data for more complex object detection, surface normal or semantic segmentation investigations. The detailed generated video sequences are categorized according to three scenarios:

**Incremental Classes:** Representing the most commonly investigated continual scenario, our video-stream consists of four video sub-sequences, each adding one distinct object class to the task. In the training set, each sub-sequence contains only one object category, where buildings $B$ in conjunction with the street-section itself are attributed to an always present 'background' class. In contrast, a separately generated test set progressively accumulates all present classes. Recall, that we can express this change throughout the video sequences with $\pi_{E,t}$. The temporal sequence of chosen Bernoulli likelihoods for the vector of buildings, trees, street-lamps, humans and vehicles is $\pi_{E,t=1} = (1, 1, 0, 0, 0)$, $\pi_{E,t=2} = (1, 0, 1, 0, 0)$, $\pi_{E,t=3} = (1, 0, 0, 1, 0)$, $\pi_{E,t=4} = (1, 0, 0, 0, 1)$. We note that after initially choosing weather and lighting conditions for $t = 0$, we further adapt their parameters to have consistent weather and lighting for the remainder of sub-sequences.

**Incremental Lighting:** The incremental lighting video sequence is based on progressive decrease in illumination intensity as a single generative factor, without adjustments to the illumination color. The generated training video stream consists of five sub-sequences, where the typically sampled uniform distribution for light intensity is collapsed to a sequence of delta distributions with concentrated mass at $\pi_{L,t=1} = 76.8$, $\pi_{L,t=2} = 19.2$, $\pi_{L,t=3} = 9.6$, $\pi_{L,t=4} = 2.4$, $\pi_{L,t=5} = 1.2$, expressed in units of Lux. The test simply consists of a growing amount of separately generated sub-sequences through time. The categorical weather distribution is parametrized to have a probability of one for clear day and all object categories exist at all times, with all locations and total amounts sampled stochastically.

**Incremental Weather:** Incremental weather video streams can be defined in terms of probabilities on the categorical weather variable. At each point in time, we set the probability for a specific outcome to 1 and all other choices to 0. The corresponding temporal sequence for occurrence of clear, rain, snow, fog, overcast conditions is then $\pi_{W,t=1} = (1, 0, 0, 0, 0)$, $\pi_{W,t=2} = (0, 1, 0, 0, 0)$, $\pi_{W,t=3} = (0, 0, 1, 0, 0)$, $\pi_{W,t=4} = (0, 0, 0, 1, 0)$, $\pi_{W,t=5} = (0, 0, 0, 0, 1)$. Probabilities for the Bernoulli variables defining the existence of object types are all set to unity, such that all objects get sampled at all times. The test video stream is defined in analogy to the incremental lighting scenario.

### 4.2 Experimental Setup and Evaluation

Each scenario is captured in a video sequence of $960 \times 540$ resolution, consisting of multiple approximately 15 minutes long sub-sequences with 150 sampled street segments, and a respective test set video. We base encoders, and the VAE decoder, on the popular four convolutional layer architecture of Radford et al. [53], without temporal dependency and thus a frame-wise prediction. In all our models we make use of a single classification head for all tasks in the presented sequence.

We presently focus on monitoring of simple classification accuracy and train the neural networks to full convergence on a sub-sequence before proceeding. That is, the train sets only ever consist of data from the current sub-sequence/time-step. In contrast, the test set accumulates data successively.

To give an example based on the class incremental scenario, the train set of task 1 thus consists of the omnipresent background class and trees, whereas task 2 consists of data featuring background and cars. In contrast, the test set accumulates observed classes and the test accuracy is measured over all classes seen up to the present time step, i.e. task 2 would classify background, trees and cars. This procedure mimics the typically conducted evaluation in class incremental learning scenarios, in which the test performance provides a rather direct indication whether the former tasks are being catastrophically forgotten in continuous optimization.

As such, the overall objective grows in complexity over time, along with the number of tasks presented. The accuracy of the investigated continual learning techniques is thus compared to the maximally achievable upper-bound accuracy, assuming the upfront presence of the entire video, and a naive continued training, where training greedily continues only on the current sub-sequence without any mechanism to prevent catastrophic interference. To provide information on the statistical deviation of all approaches, each experiment has been repeated 5 times. A detailed account of the training, it's hyperparameters and how datasets were generated can be found in the appendix. Our code is an extension of the public OCDVAE codebase [8] in combination with the Avalanche [54] continual learning library. It is available at: `https://github.com/TimmHess/OCDVAEContinualLearning`.

### 4.3 Continual Learning Results

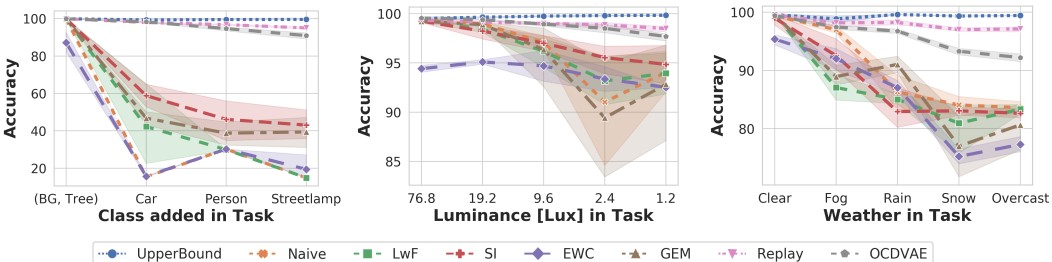

Figure 4: Comparison of deep continual learning accuracy across five experiments in three conceivable scenarios: incremental object appearance (left), decreasing illumination intensity (middle) and changing weather conditions (right). Accuracy for each method is measured after completion of the training phase for every respective task increment. The training data stems only from the current task increment to be learned, while evaluation is conducted on the cumulative test data comprising all tasks up to the present point in time. Learning without Forgetting (LwF) [9], Synaptic Intelligence (SI) [14], Elastic Weight Consolidation [13], Gradient Episodic Memory [22], direct exemplar replay, and Open-set Denoising Variational Auto-Encoder (OCDVAE) [8] are contrasted with naively continued training and the maximally obtainable accuracy when accumulating all tasks' data.

Figure 4 shows the overall achieved accuracies at the end of each sub-sequence for the considered continual learning techniques, corresponding naive continued training, and the maximally attainable upper-bound. Note that the upper-bound indicates that the three scenarios can in principle be fully solved with the chosen neural architecture. Whereas regularization methods provide some benefit in class incremental scenarios, they seem to fall behind even a naive continued training, which is supposed to incur full catastrophic interference, in the other settings. For instance, they all significantly under-perform with global weather changes. Although the employed OCDVAE generative replay can be observed to prevent the occurrence of catastrophic interference almost completely for class increments, it also begins to struggle when the environment's weather changes. We suspect that this might be due to the generative model having more difficulties in capturing the statistics of weather and its implications such as puddles, in contrast to only having to accurately generate different objects under fixed environmental conditions.

Interestingly, the trends observed in regularization techniques seem to be mirrored when using gradient episodic memory, even though it makes use of auxiliary stored data examples. We hypothesize that this is a consequence of GEM nevertheless relying on a regularization technique at its core, that is, the employed gradient constraints based on the stored pattern information. In the absence of explicit task labels this seems to be challenging. In light of this result, it is particularly interesting to observe that a more straightforward exemplar replay implementation, where the same amount of retained data instances is simply interleaved directly into the continuous training process, seems

to achieve accuracies that are sufficiently close to the upper-bound. Naturally, such an approach to continual learning could perhaps be viewed as the most trivial solution, where performance is directly proportional to the amount of retained original data instances.

### 4.3.1 Continual Learning with Quasi-illumination Invariants

Following the results of figure 4, we observe that some deep continual methods are less generically suitable than initially advocated. In practice however, we desire robustness to broader amounts of scenarios, made accesible through our simulator. As an example, mitigating performance degradation as a result of homogeneous lighting changes in the raw video through transformation into illumination invariant spaces has been well known for multiple decades [55, 56, 57]. To showcase the severity of the deep methods' shortcomings, we repeat the naive continuous training in the progressive lighting experiment with an included photometric color invariance operation.

Based on the assumption that color ratios are quasi-invariant under a dichromatic reflection model with white illumination [56], we can define: $c_1 = \arctan(R/\max\{G, B\})$, and corresponding definitions for the other two channels. Table 1 shows that such pre-processing halves the gap to the upper-bound. Note how inclusion of such a simple assumption already leads to a naive greedy deep network rivalling and even surpassing the accuracies of the continual learning specific designs. Making use of another long known invariant visual descriptor, local binary patterns [58, 59, 60, 61], the accuracy in table 1 even closely approaches a 100%, see the appendix for further details. This further highlights the importance of considering the nature of diverse scenarios, as deep continual learning should ideally leverage quasi-invariant spaces where possible to be stable.

Table 1: Incremental lighting experiment under consideration of a photometric color invariant or local binary patterns (LBP).

| Illumination Intensity [Lux] | Accuracy [%] | | |
| --- | --- | --- | --- |
| | Naive | Naive + photometric color invariant | Naive + LBP |
| 76.8 | 99.20 $\pm_{0.1}^{0.1}$ | 98.66 $\pm_{0.19}^{0.15}$ | 99.18 $\pm_{0.05}^{0.06}$ |
| 19.2 | 97.11 $\pm_{1.46}^{1.20}$ | 98.61 $\pm_{0.98}^{0.47}$ | 99.27 $\pm_{0.09}^{0.12}$ |
| 9.6 | 93.55 $\pm_{2.7}^{2.58}$ | 98.61 $\pm_{0.36}^{0.21}$ | 99.26 $\pm_{0.05}^{0.07}$ |
| 2.4 | 91.55 $\pm_{0.14}^{1.00}$ | 97.56 $\pm_{0.76}^{0.76}$ | 99.42 $\pm_{0.03}^{0.05}$ |
| 1.2 | 90.89 $\pm_{2.39}^{1.61}$ | 95.28 $\pm_{2.07}^{1.32}$ | 99.40 $\pm_{0.04}^{0.04}$ |

## 5   Discussion of Simulator Use-Cases and Prospects for Analysis

Our presented empirical investigation has consciously focused on rather simple classification tasks in an attempt to provide an initial experimental showcase for our simulator's utility. The rationale behind this choice has been two-fold: a) The experiments should be directly relatable to the community with respect to following the predominant set-ups of prior investigations, e.g. incremental MNIST, CIFAR and similar continual classification practices (even though it could be argued whether seeing only trees, or only cars is a realistic assumption in practice). b) The experiments seem to sufficiently demonstrate that the phenomenon of catastrophic interference in continual learning requires a more principled exploration in more diverse and controlled settings.

We note that, as a benefit of our flexible simulator design and its accessible modularly parametrized generative model, the presented classification experiments represent but a small subset of readily assessable future experiments. To point out the present investigation's limitations, we highlight immediately conceivable investigations in a short outline:

- **Semantic segmentation & modalities:** Continual learning investigation in semantic segmentation [62, 63, 64, 65]. On the one hand, we can conduct experiments in direct extension to the presented classification ones. On the other hand, we could investigate an alternative where objects are always present and instead labels are progressively added to become increasingly fine-grained, see the example of figure 1. Similarly, analysis can be extended through consideration of the simulator's other modalities, such as surface normals and depth.

- **Frequency of occurrence:** Our experiments have presently focused on introduction of a single class at a time or illumination and weather conditions being equally likely. In practice, it is certainly the case that probability of occurrence plays a major role. We expect that future investigations can adapt our presently assumed probabilities, for instance, in order to investigate scenarios with rare occurrences, or continual learning scenarios where concepts appear or disappear multiple times throughout the entire video sequence.

- **Identifying distribution shift:** Present continual learning mechanisms of the experiments were provided with task-boundary information. The increasingly important question of whether a devised approach can identify various sources for, potentially continuous or gradual, distribution shift can and should be considered (e.g. to decide when to learn continually or when to protect the model from catastrophic interference) .

- **Disentangled representations:** Apart from above straightforward prospects, recall that our simulation has explicitly laid open and parametrized physics-based rendering properties. This leads to multiple imaginable video sequences that we believe will have particular importance for future work. In figure 1 an example where the scene is rendered without material object normals, surface roughness and is devoid of color has already been depicted. There is no light reflections or refractions such that the image presents a simplified gray-scale world with a focus on geometry. Governing such physics-based rendering properties in conjunction with real-time control over appearing objects and environmental conditions can facilitate future analysis into the disentanglement of representations in deep generative models [66, 67, 68], enable further investigation into the debate on texture versus shape bias in deep learning [69, 70], or allow for the analysis of meaningful learning curricula of increasing complexity [71].

- **Temporal consistency:** Finally, it is worth to remember that our simulator renders temporally cohesive video streams, even though our initial classification experiments have considered frames independently. All of the above suggestions can thus be conducted under consideration of temporal consistency, arguably being a natural mode of data presentation for continually learning systems.

In addition to these mentioned prospects, we point out that future analysis could also consider the degree of transfer from trained simulated models to the real-world. Although this is not the essential premise of our work, the latter could be regarded as a present limitation of our work. As with any simulator, the degree of transfer inevitably scales with the availability of high-fidelity assets. To this end, please see the appendix, where we provide a more detailed discussion on this issue with respect to our simulator. Nevertheless, we recall that catastrophic interference should ideally first be overcome in well understood simulation before deducing generic mechanisms with opaque interpretations on real-world applications. A final assimilation of simulator to real-world statistics could thus be regarded as a subsequent goal [48, 49].

## 6   Conclusion

We have introduced a parametric interpretable generative model and its 3D graphics engine realization for the procedural online generation of continual learning scenarios. It provides a rich set of flexible generative factors that are adjustable by a straight forward configuration and cover all aspects of the continuously evolving virtual world. This allows the user to easily generate temporally consistent data streams which would require potentially insurmountable effort to be acquired in the real-world. To bootstrap the proposed benchmark generator, initial exploration on the basis of three distinct generated scenarios has been aligned to the currently employed evaluation scheme of using a set of successive tasks, each composed of an iid classification dataset. Without raising the complexity to temporally consistent online learning, or objectives such as semantic segmentation, the presented experiments already highlight the necessity of such a simulation for more extensive evaluation, in order to analyze and overcome the current shortcomings of continual deep learning mechanisms.

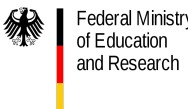 **Acknowledgements:** This work was supported by the Artificial Intelligence Systems Engineering Laboratory (AISEL) project under funding number 01IS19062, funded by the German Federal Ministry of Education and Research (BMBF) program "Einrichtung von KI-Laboren zur Qualifizierung im Rahmen von Forschungsvorhaben im Gebiet der Künstlichen Intelligenz".

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
