# OpenReview forum: "A Procedural World Generation Framework for Systematic Evaluation of Continual Learning"
_NeurIPS.cc/2021/Track/Datasets_and_Benchmarks/Round1 — NeurIPS 2021 Datasets and Benchmarks Track (Round 1)_

### Official Review · Reviewer_XSky · 2021-07-05
**A good paper for generating continual learning datasets**

**Rating:** 7
**Confidence:** 4

**Strengths:**

I think overall the contribution is significant in the continual learning community. The lack of realistic datasets has been a problem in the continual learning community for a long time, and this paper proposes one option to resolve it. The datasets and implementations are released and accessible to the general public. I did not see any ethical and social issues. Detailed comments:

1) The proposed framework is flexible. Users can use it to generate different types of video sequences (e.g. different weather condition) to evaluate continual learning algorithms in different applications. The tasks look more realistic than several commonly used datasets such as the split CIFAR and rotation MNIST.

2) In addition to classification, the dataset also supports other tasks such as image segmentation. The authors did not provide experiments on tasks other than classification, but as the authors mentioned, other tasks can be evaluated using this dataset.
Moreover, since the dataset is video, it also allows the study of the temporal relationship between the frames, which has not been explored enough in continual learning community.

**Weaknesses:**

I think the weakness of this paper is mainly in the experiment section (Section 4).

1) Although this paper proposes a more realistic data generation framework for continual learning, I think some datasets that the authors used to benchmark existing algorithms are not realistic enough. For example, the incremental class task does not seem to be realistic in the real urban driving scenario. In practice, it's unlikely that in the first video sequence we only observe trees and in the second sequence we only observe street-lamps, etc. Another suggestions is that for weather condition change, I can imagine that different weather conditions (e.g. clear vs fog) can appear with different probability and some bad condition appears less likely but are important for driving. Right now in the benchmark different weather conditions appear equally likely.

2) I am a bit confused by the results shown in Figure 4. I am not sure what the y-axis "Accuracy" mean here. Is it the accuracy over all the distributions (e.g. in the left figure, all the objects?) If that's the case, then after the first task, the algorithm has only seen trees and not other objects, how can the algorithms all have >80% accuracy (over all the objects) after the first task in the left figure?

3) I am also a bit skeptical on the results that many other methods do not outperform naive continual learning baseline. Intuitively, at least for the methods that stores replay data from previous sequences (e.g. GEM) should not be worse than the naive baseline since in the worst case they can simply discard the replay data and recover the performance of the naive algorithm, and in the literature the replay-based algorithms usually have good performance. I am wondering if the algorithms are carefully tuned or not.

**Additional Feedback:**

See the review in other sections.

**Clarity:**

The paper is overall clearly written. As mentioned in the "Correctness" section, I think it's hard to follow all the details of the generative model. In addition, as mentioned in the "Weakness" section, I think there are several places that are not clear enough in the experimental results.

Typo: Line 238 its -> it’s

**Correctness:**

I think the dataset is constructed in a sound way. The graphical model is quite complicated (see Figure 2), and I think it's a bit hard to follow all the details of this generative model. However, I do think it seems to be a reasonable model for generating urban driving videos. I have some concerns on the experimental results as mentioned in the "Weakness" section.

**Documentation:**

The dataset documentation is sufficient.

**Ethics:**

No.

**Relation To Prior Work:**

The discussion on prior work is clear and sufficient.

**Summary And Contributions:**

This paper proposes a graphical model for generating datasets for continual learning. The data are synthetic video sequences of urban scenes that can occur in driving, generated by a computer graphics framework. The users can set different configurations for the objects in the scene, the weather condition, illumination intensity, etc. The authors generated three different types of video sequences and benchmarked several existing continual learning algorithms.

---

> ### Author Response · Authors · 2021-07-09
> **Author response to reviewer XSky:**
>
> Thank you for your positive review. We agree with your presently raised weaknesses of the experimental section and will include your suggested improvements through descriptions with the addition of the extra page.
>
> 1. **Experiments potentially not realistic enough**: We agree that some of the datasets we use in our investigation could be improved in terms of more realistic composition. We will add this argument to our limitations section. As our investigation was supposed to provide an empirical introduction, we have decided in favour of initial experiments that can be immediately related to present work and are easy for the community to follow. In turn, we have opted to mirror the predominant class incremental setup that a majority of papers pursue: i.e. first seeing MNIST 0,1, then 2,3 … even though this isn’t really realistic for either handwritten digits nor the classes in our simulator. As you have correctly pointed out already, this doesn’t impair the capabilities of our simulator and future investigations. The latter was also our rationale behind adding section 5, so that the reader is fully aware that the investigation is really meant to be early stage and future works are encouraged to pursue more realistic scenarios (and also temporal consistency, semantic segmentation etc.). Similarly, we hope that future users can set different priors on the chance of weather conditions occurring.
>
>
> 2. **Clarity of experimental description & figure 4 results**:
> We will extend the experimental description to clarify how evaluation is conducted. After your raised point, we realize that this has been rather short and may be insufficient in its present state.
>
> 	As a brief summary for this response, as stated in above point 1, we presently mirror the evaluation setup of previous works:
> 	- the train set only ever consists of the data in the current time step. For example, the model is at any time of training only exposed to cars, then only persons, or e.g. one weather condition at a time (+ omnipresent background).
> 	- In contrast, the test set accumulates the data sequentially. For instance, the accuracy of task 1 would thus first be measured over classifying background vs. trees, in the next step background vs. trees vs. cars and so on. Hence, the accuracy is over all test data *observed up to this point in time*.
>
>        Again, this is a choice to resemble previous evaluations, in which “forgetting” can be gauged more easily. As such, the overall objective grows in complexity. Note however, that the difficulty doesn’t increase to the extent that the upper-bound experiences a natural decay in accuracy (i.e. if we were to also just simply accumulate time steps in the train data, as done for testing), as seen in our plots. We imagine that future investigations can also include different ways of evaluation, e.g. training on sequence parts, but always evaluating on a full test set.
>
>
> 3. **Hyper-parameters & exemplar accuracy**: Your raised point on hyper-parameters and GEM is very interesting. We had initially conducted a grid-search over most hyper-parameters on held-out validation sequences. However, particularly for regularization type approaches, we could not find a value that consistently yielded significantly more accuracy than others. We have not tuned the memory size and fixed this to a small percentage (roughly 2.5%) of the dataset, because larger memory buffers will naturally increase accuracy. We have made sure to cross-check our own implementation by reproducing our work in the PyTorch Avalanche continual learning library (we now have a pending pull request for our dataloaders/datasets to be included. This should also facilitate reproducibility.)
>
>       In terms of GEM, from our experimental logs, it looks like the challenge in GEM isn’t necessarily related to the fact that exemplars are retained, but that the update involves regularization techniques, i.e. the employed gradient penalties. This seems to be challenging when task ids are not given, which is something that the original paper assumes to be provided, but we (and many other works) consider to be a too strong assumption. We will add these details to the corresponding training details supplementary material section.
>
>
> **Clarification through additional exemplar experiments**: We are presently running experiments with exemplar rehearsal in the form of storing the raw images and simply interleaving them directly in the continued training process (e.g. as in iCarl). In the first batch of results, this seems to perform significantly better than GEM (on par with the reported generative replay). We will add these experiments to the main body. We’ll also elaborate in more detail on hyper-parameters in the appendix. Please note that we unfortunately will likely not be able to finish this update before the discussion deadline on Wednesday, as we need to properly assess the statical deviations across multiple runs/seeds.

---

> > ### Comment · Reviewer_XSky · 2021-07-13
> > **Thanks for your response**
> >
> > Thanks for your response and I think it makes sense to me. In particular, some of my confusion about Figure 4 is clarified. Overall I think it's a good contribution and I would keep my score.

---

### Official Review · Reviewer_SZ43 · 2021-07-05
**Very useful dataset tool for the community**

**Rating:** 8
**Confidence:** 3
**Correctness:** Yes, the dataset construction method …
**Clarity:** Yes.

**Strengths:**

This is exactly the kind of work that this track is meant to encourage -- very difficult to design and architect simulator, well thought-through and clearly documented. The authors have also took care to make the simulator available as an executable, thus bundling 3D assets that may be difficult to accept in their raw form. The value of raising awareness of this useful tool outweighs any possible flaws in the submission details.

**Weaknesses:**

As any handcrafted CG simulator, results obtained using this simulator may not transfer well to real-world scenarios. Yet, this is tool is a clear step forward.

**Additional Feedback:**

N/A.

**Documentation:**

Yes.

**Ethics:**

No issues.

**Relation To Prior Work:**

Yes.

**Summary And Contributions:**

This paper proposes a very useful and carefully crafted simulator to the computer vision community, in the context of autonomous driving and continual learning over dynamic content.

---

> ### Author Response · Authors · 2021-07-09
> **Author response to reviewer SZ43**
>
> Thank you for your very positive review. We appreciate that you agree with us on NeurIPS benchmarks & datasets being an ideal venue for these types of works.
>
> As there isn’t any major concerns raised in this review, we keep our response short. We agree with your raised weakness on CG simulators having their limits. We also fully agree that our tool is still an important first step and are excited for its use and potential future extensions. We invite you to also read the responses to the other reviewers, or participate in the respective discussion.

---

### Official Review · Reviewer_kGbg · 2021-07-05
**Interesting use of simulators to evaluate continual learning**

**Rating:** 6
**Confidence:** 4
**Correctness:** The paper appears to be correct.

**Strengths:**

The idea of using simulators to evaluate synthetic learning might provide a systematic way of evaluating approaches as good benchmarks are lacking for this task. Since the generative model is controllable in interpretable ways, analysis can be done along different dimensions providing more insight into ways to improve continual learning. Some of these aspects are illustrated in their study on model's robustness to new classes and weather conditions.



**Weaknesses:**

Similators have their own limitations due to the narrow range of visual appearances and other shortcuts employed by graphics engines (e.g., coarse geometry but detailed texture) for realism. This is also a reason for skeptisism for the proposed conjecture. In particular, it would be useful to characterize what good performance on this dataset tells us about perhaps a real-world use case.



**Additional Feedback:**

None.

**Clarity:**

The clarity can be improved. Several pages in the paper appear as a block of text, which could be shorted into bullet points. It would help to include some figure to illustrate the variety of the dataset and the types of mistakes the models makes in the main paper.

**Documentation:**

Yes.

**Ethics:**

Not that I can think of.

**Relation To Prior Work:**

Prior work is well discussed.

**Summary And Contributions:**

The paper makes two contributions.

1. It presents a simulator to generate synthetic visual data. Various parameters such as the number of classes, lighting and weather conditions, can be changed procedurally to introduce domain shifts across time.
2. It presents an analysis of several continual learning approaches as novel classes are introduced, etc.

---

> ### Author Response · Authors · 2021-07-09
> **Author response to reviewer kGbg**
>
> We thank you for your positive review. We are in agreement with the limitations you pointed out.  Our hope is that our proposed work can serve as a first step in the right direction of broader and systematic evaluation in continual learning.
>
> **Simulator limitations and real-world transfer**
> While simulation is indeed limited and there is a gap between simulations and real data, we believe that the contributions in their present form provide a systematic way of assessing progress in the continual learning community. Present experimental methodology for evaluation is still limited - i.e. incremental or permuted MNIST/CIFAR datasets. The community is thus in need of having tools for more controlled and systematic ways towards rigorous experimentation. Providing one such tool is the main objective of our work. Our focus lies on investigation of generic continual learning phenomena, such as catastrophic interference. Developing real world ML applications by learning on simulation and bridging the gap between sim-to-real is not the focus of our paper.
>
> Although we fully agree that sim-to-real transfer is an important research question,  we have not added experiments in this direction for two additional reasons:
>
> 1. As catastrophic interference is central to continual learning, even in simulation, we think that it is beneficial to explore implications in controllable environments with various configurations. These are typically very difficult to realize when gathering real datasets. We are unaware of any real open datasets, apart from knowledge of proprietary data sets in industry, that would easily allow to permute the order or control the extent of weather condition changes, lighting variations, control which objects are observable, or even just change of one factor at a time in a meaningful way. We agree that further modification is needed to our present approach in order to allow sim-to-real transfer.
>
> 2. There is rich literature, some of which is referenced in our related work section, which aims to address the gap between sim-to-real, e.g. several works on employing GANs to adapt simulated statistics to match real world data or frameworks to tune simulator parameters directly. We believe that delving extensively into these techniques to further match simulated with real world statistics would be out-of-scope of the present contribution. In particular, our experimental results already show that it is quite challenging to even tackle catastrophic interference in virtual worlds as is.
>
> Since we are allowed to add another page of text in our paper, we will modify our paper to further emphasize our focus and limitations of the present approach.
>
> **Model mistakes:**
> We understand the motivation behind your comment on the value of adding images of potential mistake cases. Indeed, in many applications this can provide valuable additional insights to the reader. However, in the particular continual learning case, most of the made mistakes are of catastrophic nature. That is, for instance, when progressing along the task sequence, many of the current methods are not able to maintain old task knowledge (or if they maintain old knowledge, cannot adequately encode new concepts). As such, e.g. a formerly correctly classified tree, will now simply be attributed with a high confidence of belonging to any last-learned task label.
>
> **Figure/Video**:
> The pdf format is a little constrained to showcase the simulator capabilities/modes through multiple figures. We will make an extension to figure 1 to showcase more images. In addition, we have composed a continuous 4 minute video sequence from the simulator, where sub-sequences are sampled such that potential use of on-the-fly abrupt/continuous changes is illustrated. For the purpose of the review, it is uploaded here: [https://hessenbox-a10.rz.uni-frankfurt.de/getlink/fiQwmoWzKgJeDWnzizscZJ4t/CL_Simulator.mp4](https://hessenbox-a10.rz.uni-frankfurt.de/getlink/fiQwmoWzKgJeDWnzizscZJ4t/CL_Simulator.mp4). We will make an update with uploaded video resources to our repositories as well and include links inside the paper. This should give the reader a better impression than static images.
>
> **Clarity:**
> We have tried to introduce as many paragraphs, subsections and itemizations as possible. We understand, that some sections, like 2.2 nevertheless largely consist of multiple paragraphs in sequence. We believe this can be improved by introducing small subsection headings that summarize the scope of the paragraph (e.g. categorical variables, dynamic actors etc.). Other than that, we would appreciate if you could point us to more specific sections where blocks of texts need improvement.

---

### Author Response · Authors · 2021-07-09
**Comment for all reviewers. Initial response has been posted**

Dear reviewers,

We appreciate all the reviewers' efforts and provide responses to each individual reviewer.

Our comments mainly involve minor clarifications and our proposed revisions based on your feedback to strengthen the contribution of our paper.

Please note that we did not immediately update the pdf with the extra page just yet, as some of these updates will require more time. For instance, the exemplar rehearsal experiments will require their time to actually be conducted in all scenarios with statistical deviations across experimental repetitions. We are working very hard on this but are not sure this can be finished before next Wednesday. However, we guarantee that all our described updates will be ready for a camera ready version of the paper.

Thank you again for the very positive reviews. We are looking forward to the discussions!

---

### Author Response · Authors · 2021-07-13
**Uploaded revised pdf, incorporating reviewer feedback and improvements outlined in responses**

Dear reviewers,

Thank you again for your reviews. We have uploaded a revised version of our main paper. We have used the additional page for improvements based on your feedback. We summarise our additions briefly in this comment. They correspond to what has been written in the detailed responses to the reviewers. Specifically, we have:

* Improved readability/structure in terms of having less blocks of text. We have introduced multiple paragraphs, small headings in section 2.2, have separated the experimental description chunk into separate subsections for general description, investigated scenarios, set-up and evaluation, have introduced a small heading for the illumination quasi-invariants experiments.
* We have revised the experimental descriptions significantly. There now should be a clear description of how train and test accuracy are measured, both in terms of the “experimental setup and evaluation” paragraph, as well as figure 4 caption.
* We have added exemplar replay as a method to our experimental section, to have a method in direct comparison with GEM.
* We have extended the results discussion, including a small discussion on GEM and exemplar rehearsal.
* We have added a small paragraph at the beginning of section 5 to more clearly emphasise our experiment’s intention + limitations, before heading into prospects. This should complement the limitation paragraph at the end of section 5 with respect to future work in terms of transfer from simulated to real world.
* We have added a bullet point on “frequency of occurrence” in the discussion section to point out the limitation/prospect of conducting experiments where probabilities are adapted for concepts to not appear with equal likelihood (e.g. weather).
* Extended figure 1 to give a more complete overview.  Please also note the video we had already posted in the responses to provide a better intuition (https://hessenbox-a10.rz.uni-frankfurt.de/getlink/fiQwmoWzKgJeDWnzizscZJ4t/CL_Simulator.mp4). We are going to upload this to one of the project’s repositories for readers/users to view.

---

### Decision · Program_Chairs · 2021-07-26

**Decision:**

Accept

**Comment:**

This paper proposes a well-designed and well-documented simulator to help study continual learning. Like all reviewers I recommend acceptance.